# Identification of ROH Islands Conserved through Generations in Pigs Belonging to the Nero Lucano Breed

**DOI:** 10.3390/genes14071503

**Published:** 2023-07-23

**Authors:** Paola Di Gregorio, Annamaria Perna, Adriana Di Trana, Andrea Rando

**Affiliations:** Scuola di Scienze Agrarie, Forestali, Alimentari ed Ambientali, University of Basilicata, Via dell’Ateneo Lucano 10, 85100 Potenza, Italy; anna.perna@unibas.it (A.P.); adriana.ditrana@unibas.it (A.D.T.); andrea.rando@unibas.it (A.R.)

**Keywords:** Nero Lucano pig, Southern Italy, inbreeding coefficient (F_ROH_), runs of homozygosity (ROH) islands

## Abstract

The recovery of Nero Lucano (NL) pigs in the Basilicata region (Southern Italy) started in 2001 with the collaboration of several public authorities in order to preserve native breeds that can play a significant economic role both due to their remarkable ability to adapt to difficult environments and the value of typical products from their area of origin. In this study, by using the Illumina Porcine SNP60 BeadChip, we compared the genetic structures of NL pigs reared in a single farm in two different periods separated by a time interval corresponding to at least three generations. The results showed an increase in the percentage of polymorphic loci, a decrease in the inbreeding coefficient calculated according to ROH genome coverage (F_ROH_), a reduction in the number of ROH longer than 16 Mb and an increase in ROH with a length between 2 and 4 Mb, highlighting a picture of improved genetic variability. In addition, ROH island analysis in the two groups allowed us to identify five conserved regions, located on chromosomes 1, 4, 8, 14 and 15, containing genes involved in biological processes affecting immune response, reproduction and production traits. Only the conserved ROH island on chromosome 14 contains markers which, according to the literature, are associated with QTLs affecting thoracic vertebra number, teat number, gestation length, age at puberty and mean platelet volume.

## 1. Introduction

Farms rearing Nero Lucano (NL) pigs, a typical autochthonous black pig, are characterized by a very low level of environmental impact [1], since animals are raised outdoors in the Basilicata region (Southern Italy) and are able to exploit marginal areas and feed resources available in the environment. The sustainability of the production chain and the specific organoleptic properties of NL pig cured products are particularly appreciated by customers and, therefore, economically advantageous for breeders. In addition, the enhancement and protection of this breed is useful for the conservation of biodiversity and for the reduction in the environmental impact and climate footprint. These matters are goals of the “Farm to Fork Strategy” of the European Union [2], aiming to obtain a sustainable and affordable food chain effective for consumers, producers, climate and environment.

Starting from 2001, the NL pig breed was subject to an intervention for its recovery from extinction thanks to public authorities (the Basilicata Region, the University of Basilicata, the Regional Breeders Association, the Comunità Montana Medio Basento). However, it still suffers all the problems (for example, slow growth speed and a low number of newborns per delivery) determined by the inbreeding associated with the low number of sires and dams typical of “small” populations [3,4]. In addition, this breed is characterized by very low frequencies of alleles at the IGF2, MC4R and VRTN loci that, according to the literature, are associated with positive effects on some meat production traits in cosmopolitan breeds, and is free from malignant hyperthermia (MH) [5]. Recently, we analyzed the genotypes using the 61,565 SNPs of the Illumina Porcine SNP60 BeadChip of about 70% of sires and dams born from 2004 to 2014 of the NL pig population, in order to obtain a first picture of its genetic structure [6]. As expected for a recovering population starting from only six subjects, the analyzed individuals were characterized by high levels of inbreeding coefficients (both F_MOL_, calculated by referring to allelic frequencies, and F_ROH_, calculated according to the ROH extension), low effective population size and long generation intervals. These results depict a population still at risk and the need for actions to avoid an excessive inbreeding coefficient increase. 

At the end of 2021, we received from the owners of a farm, whose NL pigs were analyzed ten years ago, blood samples collected by the veterinary service during the normal activities of animal controls. As a consequence, we had the opportunity to make comparisons between two groups of NL pigs reared in the same farm and separated by a time period of about ten years, corresponding to at least three generations, in order to obtain a picture of the evolution of their genetic structure by using the ROH approach and to identify ROH islands characterized by a conserved structure through generations.

## 2. Materials and Methods

### 2.1. Animals

Animal blood samples were obtained from 76 Nero Lucano pigs reared in a single farm where, about ten years ago, 66 samples of the same breed had already been collected.

### 2.2. DNA Analyses

DNA samples were genotyped with the Illumina Porcine SNP60 BeadChip v2. The data quality control, accomplished by using PLINK v.1.9 [7], determined the removal of 3 samples due to the genotyping rate being lower than 95% and 2548 SNPs due to a call rate of lower than 95%. Hardy–Weinberg equilibrium was calculated by considering only polymorphic loci located on the 18 autosomal chromosomes. The runs of homozygosity (ROH) were obtained by defining a sliding ’window’ of 50 SNPs, allowing a maximum of one heterozygote and one missing call in the ‘window’ and at least 50 SNPs per ‘window’. Individual inbreeding values based on ROH (F_ROH_) were calculated as F_ROH_ = ΣL_ROH_/L, where ΣL_ROH_ is the total ROH length per individual and L is the autosomal genome length (2265.77 Mb, according to Sscrofa 11.1 Genome Assembly).

Gene location was accomplished by referring to NCBI Sus scrofa Annotation Release 106 (https://www.ncbi.nlm.nih.gov/genome/annotation_euk/Sus_scrofa/106/, accessed on 30 December 2022).

Gene Ontology (GO) enrichment analysis was performed by using DAVID Knowledgebase v2022q4 [8,9] (https://david.ncifcrf.gov/home.jsp, accessed on 15 February 2023) and protein–protein interaction by using STRING 11.5 software [10] (https://string-db.org/, accessed on 15 February 2023). REVIGO 1.8.1 software was used to reduce and visualize GO terms [11] (http://revigo.irb.hr/, accessed on 15 February 2023).

Significance tests and graphic plots were obtained by using R-4.1.2 software [12].

## 3. Results

The variation of the genetic structure of Nero Lucano pigs in a time period corresponding to at least three generations was analyzed by comparing 66 samples, NL-A, collected about 10 years ago and 73 samples, NL-B, collected in 2021 from the same farm. Samples belonging to the two groups were compared without considering the separation of individuals according to their generation since. Unfortunately, pedigree data were not available for samples collected in 2021.

The comparison of the minimum allele frequency (MAF) of the SNPs in the two groups showed that the percentage of the polymorphic SNPs increased from 67% to 83% in about ten years (Figure 1) (see also Appendix A for MAF chromosomal distribution).

The analysis of genotype distributions accomplished only for the SNPs located in the 18 autosomal chromosomes showed that in NL-B pigs, 6.32% of SNPs were not in Hardy–Weinberg equilibrium, and in 65.48% of cases, the disequilibrium was determined by an excess of homozygotes. The corresponding values in NL-A pigs were 3.43% and 46.61%, respectively.

The ROH analysis of the 73 NL-B pigs allowed us to identify 3690 total ROH, covering 28.11% of the 18 autosomal chromosomes, whereas in the 66 NL-A pigs, the number of total ROH was 3626, covering 38.63% of the autosomal genome (Table 1).

In addition, the mean ROH number per pig was 50.55 ± 10.97 in NL-B and 54.94 ± 7.66 in NL-A (t(137) = 2.7, *p* = 0.008), with a total ROH length per pig spanning from a minimum of 38.09 Mb to a maximum of 1260.11 Mb (mean 636.80 ± 219.04 Mb) in NL-B and from a minimum of 236.82 Mb to a maximum of 1366.99 Mb (mean 875.1 ± 198.49 Mb) in NL-A (t(137) = 6.7, *p* < 0.00001). As a consequence, the mean ROH length per pig was 12.47 ± 3.69 Mb in NL-B and 15.90 ± 3.10 Mb in NL-A. In both groups, the distribution of ROH among the size classes was balanced, with the exception of those with a length lower than 2 Mb. As shown in Table 1, the ROH number and genome coverage percentage showed the greatest increase for the class 2–4 Mb and the greatest decrease for the class >16 Mb in NL-B pigs.

Individual inbreeding values based on ROH extension (F_ROH_) ranged from a minimum of 0.02 to a maximum of 0.56, with a mean value of 0.28 ± 0.10 in NL-B, whereas in NL-A, the corresponding values were 0.10, 0.60 and 0.39 ± 0.09, respectively (Figure 2). As a consequence, the F_ROH_ mean value decreased by 28% in about ten years.

The number of core ROH, defined as the consensus regions determined by the overlapping of individual ROH [7], was 587 in NL-B and 494 in NL-A (Appendix A, Appendix A). ROH islands were obtained by considering an uninterrupted stretch of at least three SNPs both located in a core ROH and exceeding 99% of the standardized distribution [13] (Figure 3). By using this approach, 24 and 19 ROH islands were identified in NL-B and NL-A, respectively (Appendix A). The higher number of ROH islands observed in NL-B was, however, associated with a lower total extension (NL-B 22.79 Mb versus NL-A 29.94 Mb) (Appendix A).

As shown in Table 2, only in five ROH islands, located in chromosomes 1, 4, 8, 14, and 15, can a partial or total overlap be observed between the two groups.

According to Sus scrofa 11.1 Genome Assembly, the five overlapping regions contain 52 genes (Appendix A). Gene Ontology (GO) analysis using DAVID Knowledgebase v2022q4 showed that 26 genes were significantly involved in 15 biological processes (Table 3) whose GO terms were grouped into four superclusters (inflammatory response, system development, regulation of cellular component organization and glycerolipid metabolism) by using REVIGO software (Figure 4 and Appendix A). 

Furthermore, three genes (S100A7, S100A8, S100A9) were involved in the IL-17 signaling pathway, which is engaged in several immune regulatory functions such as the response to injury, physiological stress, and infection [14].

Analysis of the 52 genes with STRING 11.5 software evidenced more protein–protein interactions than expected (26 edges rather than 4, PPI enrichment *p*-value 8.5 × 10^−13^) (Appendix A). As a consequence, some of the proteins coded by these genes are at least partially biologically connected. However, only part of the biological processes identified by DAVID software found a correspondence with protein–protein interactions highlighted by STRING software.

According to the Pig QTL database (version Sus scrofa 11.1 Genome Assembly), only the conserved ROH island on chromosome 14 contained six QTLs affecting thoracic vertebra number (#64660 and #64771) [15], teat number (#126628) [16], gestation length (#173177) [17], mean platelet volume (#37844) [18], and age at puberty (#22109) [19] (Appendix A). Each of the six SNPs located at the peak of association with the above-mentioned QTLs were characterized by a MAF = 0 in both NL-A and NL-B pigs.

## 4. Discussion

In this work, we compared the results of Illumina PorcineSNP60 BeadChip genotyping accomplished on two groups of Nero Lucano pigs, NL-A and NL-B, reared in the same herd and sampled in two time periods separated by about ten years, corresponding to at least three generations.

Analysis of the results showed that in this period the number of polymorphic loci increased by about 24%. In addition, the ROH analyses in the two groups showed that the mean ROH number, mean total ROH length and mean ROH length per pig were characterized by decreased values in NL-B. The distribution of the ROH per class was consistent with these results, showing a reduction in the number of ROH longer than 16 Mb and an increase in the number of ROH with a length between 2 and 4 Mb in NL-B. All the above-mentioned results were responsible for the strong decrease in the inbreeding coefficient over ten years, from 0.39 to 0.28, based on ROH genome coverage (F_ROH_). The origin of this decrease could be explained either by introgression events or by the use of sires and dams belonging to other NL herds. However, according to what was stated by the breeder and to the typical morphological traits characterizing the NL-B individuals, the former hypothesis can be excluded. 

According to several authors, the frequency and extension of ROH are affected by ancient or recent inbreeding events in the population and/or by the presence of genes under strong selection [20,21]. ROH islands are stretches of homozygous genomic sequences characterized by a very high frequency in a population. As a consequence, the analysis of ROH islands has been the object of several studies in different species in order to identify genes associated with domestication or useful for the improvement of animal production and reproduction. We identified five ROH islands in common between NL-A and NL-B groups—that is, genomic regions constantly maintained through generations as homozygous milestones. Of course, genes contained in these homozygous milestones should be strong candidates to have effects on the adaptive selection of this breed. 

Some of the genes identified in these five regions are involved in biological processes affecting immune response, reproduction and production traits. For example, three of the calcium ion binding proteins belonging to the S100 family (S100A7, S100A8, S100A9) are involved in the IL-17 signaling pathway, which is important in the immunity to pathogens, or contribute to the pathogenesis of inflammatory diseases [14]. Furthermore, S100A6 is involved in the response to the viral infection causing porcine reproductive and respiratory syndrome (PRRS), one of the most economically significant swine infectious diseases [22,23]; PPGRP-S has a role in the immunity against intestinal microorganisms [24]; RHBDD3 suppresses, in mice, the production of IL6, preventing the development of autoimmune diseases [25]; and PLA2G3 affects the maturation and function of mast cells, key players in the inflammatory response [26]. The conservation of regions containing genes affecting immune response is probably due to the need for an efficient immunological system in animals living outdoors in semi-wild conditions, such as NL pigs.

As far as pig reproduction traits are concerned, the MORC2 gene prevents apoptosis of granulosa cells [27], whereas SMTN, coding for a structural protein found exclusively in contractile smooth muscle cells, is associated with ovarian follicle growth and development [28]. Furthermore, both OSBP2 and LIMK2 are involved in spermatogenesis [29]. In mice, the defective gene MORC2b is responsible for male and female sterility [30], and mice deficient in OSBP2 produce a severely reduced number of spermatozoa that show very low motility and no fertilizing ability in vitro [31]. Similar results are observed for mice with a disrupted LIMK2 gene in which the progression of spermatogenesis is strongly affected [32] and for humans where mutations in the heterozygous state in LIMK2 are present in infertile males [33]. Finally, an LIF gene polymorphism (rs3463076786: C/T) is associated with the number of stillbirths in pigs [34].

In Chinese Anqing six-end-white pig, SELENOM has been identified as a candidate gene affecting body weight [29]. This gene is highly expressed in the brain and may be involved in neurodegenerative disorders. Transgenic mice with targeted deletion of this gene are characterized by obesity, suggesting a possible role of SELENOM in the regulation of body weight and energy metabolism [35]. In addition, DOCK10 is a candidate gene associated with intramuscular fat [36], and EWSR1 is associated with meat quality traits, in particular meat colour [37]. The PLA2G3 gene belongs to the phospholipase A2 family that in pigs, as in other vertebrates, is involved in lipid metabolism [38]. In humans, KREMEN1 and ZNRF3 are involved in body fat distribution [39]. 

The conserved five ROH islands were also analyzed for the presence of QTLs, and only the one on chromosome 14 contained markers associated with effects on thoracic vertebra number, teat number, gestation length, mean platelet volume and age at puberty. Unfortunately, these markers were monomorphic in all of the genotyped NL pigs, and therefore cannot be used to analyze the variation of these traits in this breed. In any case, the conserved ROH island located on chromosome 14 could have had a key role in pig domestication since it was also identified in other breeds. In fact, the upstream region containing the AP1B1, EWSR1, KREMEN1, NEFH, THOC5, and ZNRF3 genes (see Appendix A) was identified in Russian Large White pigs as a signature of selection associated with QTLs affecting reproduction and production traits [40], whereas the downstream region containing MORC2, SMTN, INPP5J, PLA2G3 and RNF185 genes (see Appendix A) was identified by Li et al. [41] as a signature of selection in Chinese pigs. This region is associated with a QTL affecting intramuscular fat linoleic acid content [42,43] that is positively correlated with pork flavor [44]. As a consequence, it is plausible that genes contained in this ROH island could also play an important role in the quality of NL pig cured meat products which are particularly appreciated for their organoleptic properties.

In conclusion, the search for ROH islands conserved through generations can be used to mark boundaries of genomic regions to be analyzed for the identification of mutations affecting the expression of economically important genes and/or differentiating cosmopolitan from small autochthonous populations.

## Figures and Tables

**Figure 1 genes-14-01503-f001:**
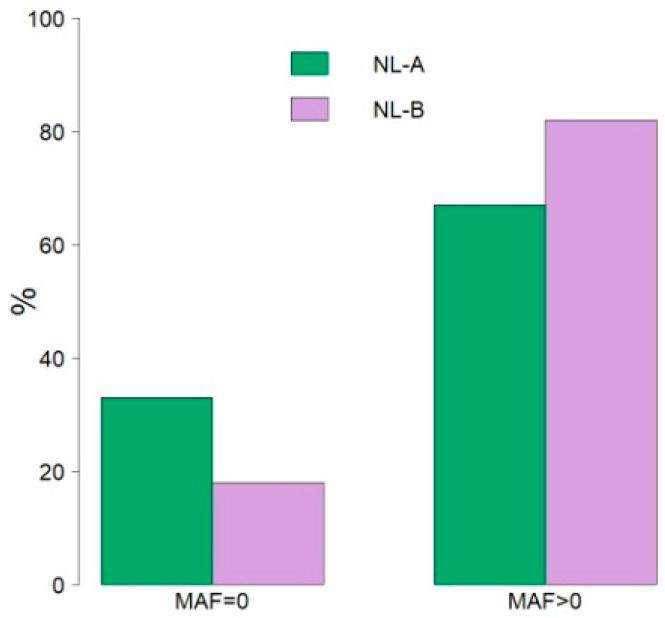
Comparison of the SNPs’ distribution according to the minimum allele frequencies (MAF) in the two Nero Lucano pig groups (z = 61.22, *p* < 0.00001).

**Figure 2 genes-14-01503-f002:**
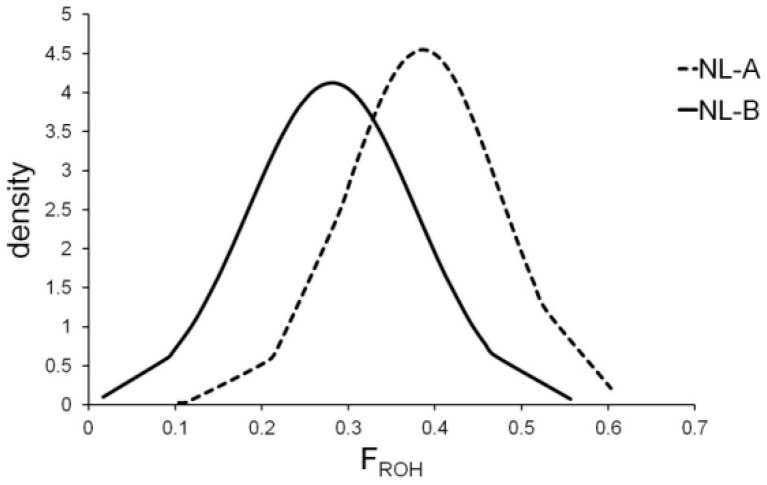
Distribution of the individual F_ROH_ (inbreeding coefficient based on runs of homozygosity) values in two groups of Nero Lucano pigs (t(137) = 6.7, *p* < 0.00001).

**Figure 3 genes-14-01503-f003:**
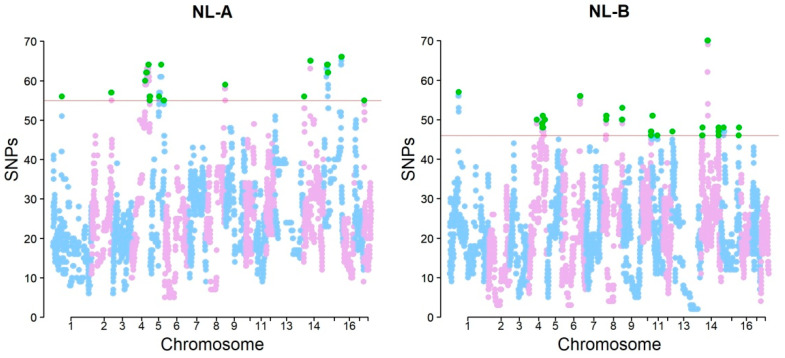
Manhattan plot of SNPs’ distribution in runs of homozygosity in NL-A and NL-B pig autosomal chromosomes. The red lines, different for each group, indicate the threshold of the top 1% of the observations. Green dots represent SNPs located in ROH islands.

**Figure 4 genes-14-01503-f004:**
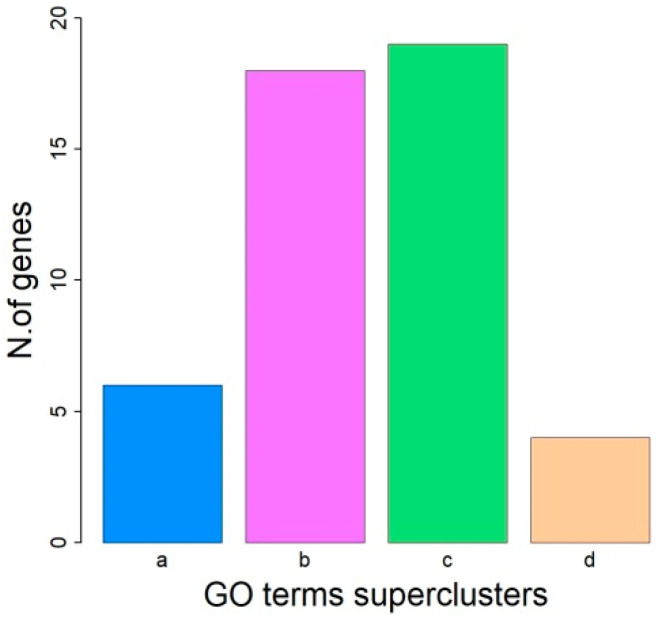
GO terms grouped into four superclusters by using REVIGO software: (a) inflammatory response, (b) system development, (c) regulation of cellular component organization (d) glycerolipid metabolism.

**Table 1 genes-14-01503-t001:** Features of ROH clustered according to length in NL-A and NL-B pigs.

ROH Class	NL-A			NL-B		
	ROH Number	SNPs/ROH (Mean ± SD)	% Genome Coverage	ROH Number	SNPs/ROH (Mean ± SD)	% Genome Coverage
<2 Mb	260	57.30 ± 7.67	0.29	236	59.82 ± 8.71	0.25
2–4 Mb	897	85.67 ± 22.93	1.73	1052	84.14 ± 20.18	1.85
4–8 Mb	859	169.26 ± 46.37	3.24	913	161.81 ± 41.76	3.16
8–16 Mb	707	329.64 ± 76.12	5.40	741	321.53 ± 72.55	5.07
>16 Mb	903	1180.92 ± 783.82	27.97	748	1002.98 ± 636.18	17.78
Total	3626	-	38.63	3690	-	28.11

**Table 2 genes-14-01503-t002:** Features (extension, number of SNPs and genes) of ROH islands conserved between NL-A and NL-B pigs.

SSC	NL-A		NL-B		N. Genes
	from-to	N. SNPs	from-to	N. SNPs	
1	61,286,411-63,123,772	52	61,286,411-63,012,075	47	1
4	96,094,167-96,682,751	19	96,094,167-96,243,742	8	7
8	135,852,700-136,018,155	4	135,852,700-136,018,155	4	2
14	46,176,964-48,062,822	60	46,176,964-47,999,414	58	39
15	126,097,384-126,642,052	17	126,097,384-126,642,052	17	3

**Table 3 genes-14-01503-t003:** Genes associated in biological processes after Gene Ontology (GO) analysis by using DAVID v2022q4 software.

GO	Biological Process	Genes
GO:0006954	inflammatory response	S100A12, S100A8, S100A9, OSM, PLA2G3, RHBDD3,
GO:0048731	system development	LIF, LIMK2, NF2, S100A8, S100A9, THOC5, AP1B1, CUL3, GAL3ST1, GAS2L1, INPP5J, KREMEN1, NEFH, PLA2G3, RHBDD3, SELENOM, ZNRF3
GO:0051128	regulation of cellular component organization	LIF, LIMK2, MORC2, NF2, S100A9, CUL3, GAS2L1, HNRNPD, INPP5J, KREMEN1, MTMR3, PLA2G3,
GO:0033043	regulation of organelle organization	LIF, LIMK2, MORC2, NF2, CUL3, GAS2L1, HNRNPD, MTMR3,
GO:0009914	hormone transport	LIF, OSM, PLA2G3, SELENOM,
GO:0051046	regulation of secretion	LIF, S100A8, OSM, PLA2G3, RHBDD3,
GO:0046903	secretion	LIF, S100A8, OSM, PLA2G3, RHBDD3, SELENOM,
GO:0050729	positive regulation of inflammatory response	S100A8, OSM, PLA2G3
GO:0031345	negative regulation of cell projection organization	LIMK2, INPP5J, KREMEN1,
GO:0023061	signal release	LIF, OSM, PLA2G3, SELENOM,
GO:0051247	positive regulation of protein metabolic process	LIF, LIMK2, S100A8, TBC1D10A, CUL3, HNRNPD, OSM, RHBDD3,
GO:0010648	negative regulation of cell communication	LIF, NF2, CUL3, CASTOR1, KREMEN1, OSM, ZNRF3,
GO:0023057	negative regulation of signaling	LIF, NF2, CUL3, CASTOR1, KREMEN1, OSM, ZNRF3,
GO:0048232	male gamete generation	LIMK2, GAL3ST1, OSBP2, PLA2G3,
GO:0046486	glycerolipid metabolic process	GAL3ST1, INPP5J, MTMR3, PLA2G3,

## Data Availability

The data analyzed during the current study are available from the corresponding author on reasonable request.

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
