# Peer review of "Identification of ROH Islands Conserved through Generations in Pigs Belonging to the Nero Lucano Breed"

_genes, 2023, doi:10.3390/genes14071503_

Round 1
Reviewer 1 Report
Dear authors,
I find the submitted paper well written, concise, with good potential for publishing. However, with several concerns:
The study was done based on data from just one farm. This is not a problem per se, however, the results should be placed in the context of the whole population, if possible.
Line 41-43 it is clear that inbreeding can affect reproductive and productive traits. However, without exact analysis for particular breed, it is not clear is the low performance results of inbreeding or is it just the genetic basis, as in other local breeds of pigs. Are there any studies for the Nero Lucano breed that can confirm this?
Line 178 - the explanation about using sires and dams from other herds seems to be logical; however, since the study has obtained results on only one farm, this should be placed in the context of the whole population: I am not completely familiar with analyzed breed, however, I think it would be useful to include information about whole population variability, such as Valluzzi et al (2021) or similar. Moreover, is it possible that decrease in ROH number is the consequence of using crossed boars, or boars of other breeds? This seems to be logic, especially if the genetic diversity in the whole population is low. I find this an important thing to resolve in the Discussion chapter.
Author Response
- We know that data were obtained from just one farm. However, this is a communication aiming to identify ROH islands conserved through generations. The paper demonstrates that such identification is possible and, hopefully, the extension of this analysis to the whole population and to other populations will be done in the next future. Compatibly with availability of funds.
- No affordable data on the performances of this breed are actually available. However, breeders always complain that these animals have few piglets per delivery (4-6) and grow slowly since, compared with cosmopolitan breeds, it takes from 3 to 5 months more to reach the slaughter weight. Furthermore, in our paper of 2021 it is clearly stated that this breed was rescued from only 6 individuals repeatedly mated among them in order to obtain a certain number of individual to be distributed to 13-14 herds spread in the Basilicata region. This means that for sure inbreeding affects the performances of this breed.
- The answer to the first part of your observation has been given at point 1. As far as we know, as stated in the paper, the breeder followed the advice on the use of sires and dams belonging to the SNL breed obtained from other herds. We are sure that not so many other breeders followed this advice. In any case we modified lines 187-191.
Reviewer 2 Report
Comments are below:
1. Relationship (either genetic or pedigree) of 73 NL-B and 66 NL-A pigs should be described.
2. In figures 1 and 2, significant test is needed.
3. Lines 107-108, significant test is also needed.
4. If pedigree information is available, how is inbreeding coefficient based on pedigree when compared to ROH-based?
5. How inbreeding coefficient in NL pigs compared to other pig populations previously reported?
6. The manuscript highlighted 5 conserved ROH regions, but more interestingly in my opinion, should also focus on different ROH regions between populations. Those different ROH regions might explain why inbreeding coefficient decreases in past ten years. Moreover, whether those ROH regions are related to the increase of the number of polymorphic loci observed in figure 1.
7. Discussion should more focus on interpreting why the number of polymorphic loci increases? It is because of the “Intervention for the recovery from 39 extinction”? How the population is managed in the past years, and whether there are human-mediated mating system?
8. As conserved ROH regions were identified, how these regions are compared to other populations? I guess some of ROH regions are results from domestication, but some are from recent inbreeding event.
9. The different ROH regions/island between population might reflect human intervention in the past ten years, deserved to be discussed as well in my opinion?
Author Response
- Done: see lines 92-94
- Done
- Done: see lines 113 and 116
- Pedigree information not available. See lines 92-94
- This is a communication aiming to identify ROH islands conserved through generations. The extension of this analysis to other population will be done in the next future. Compatibly with availability of funds.
- The 5 conserved ROH regions are ROH islands, that is, ROH with specific characteristics (lines 194-195). Since FROH is calculated as FROH = ΣLROH/L, where ΣLROH is the total ROH length per individual and L is the autosomal genome length (lines 76-78) different ROHs with the same extension will give the same FROH. The increase in polymorphic loci is responsible for the decrease in the number and length of DNA segments containing stretches of homozygous loci in each individual with the consequent decrease of inbreeding coefficient (lines 180-187)
- As stated in the introduction, animals are raised outdoors, free in the environment. The only activity of the breeder to control reproduction refers to change boars and sows. We are absolutely far from the typical industrial swine herd in which sow X is brought to the boar Y for mating. See also lines187-191
- See reply to number 5
- See reply to number 7